# Role of HO-1 against Saturated Fatty Acid-Induced Oxidative Stress in Hepatocytes

**DOI:** 10.3390/nu13030993

**Published:** 2021-03-19

**Authors:** Noriyoshi Ogino, Koichiro Miyagawa, Kenjiro Nagaoka, Yuki Matsuura-Harada, Shihona Ogino, Masashi Kusanaga, Shinji Oe, Yuichi Honma, Masaru Harada, Masamitsu Eitoku, Narufumi Suganuma, Keiki Ogino

**Affiliations:** 1Department of Environmental Medicine, Kochi Medical School, Kohasu, Oko-cho, Nankoku City, Kochi 783-8505, Japan; n-ogino@med.uoeh-u.ac.jp (N.O.); shihonam@gmail.com (S.O.); meitoku@kochi-u.ac.jp (M.E.); nsuganuma@kochi-u.ac.jp (N.S.); 2Third Department of Internal Medicine, School of Medicine, University of Occupational and Environmental Health, Iseigaoka 1-1, Yahatanishi-ku, Kitakyushu 807-8555, Japan; koichiro@med.uoeh-u.ac.jp (K.M.); masashik@med.uoeh-u.ac.jp (M.K.); ooes@med.uoeh-u.ac.jp (S.O.); y-homma@med.uoeh-u.ac.jp (Y.H.); msrharada@med.uoeh-u.ac.jp (M.H.); 3Laboratory of Hygienic Chemistry, College of Pharmaceutical Sciences, Matsuyama University, Matsuyama, Ehime 790-8578, Japan; k.nagaoka@g.matsuyama-u.ac.jp; 4Department of Biofunction Imaging Analysis, Graduate School of Medicine, Dentistry, and Pharmaceutical Sciences, Okayama University,1-1-1 Tsushima naka, Kita-ku, Okayama 7008530, Japan; ph422132@s.okayama-u.ac.jp

**Keywords:** palmitic acid, lipotoxicity, fatty liver, oxidative stress, hem oxygenase-1

## Abstract

Increased circulating levels of free fatty acids, especially saturated ones, are involved in disease progression in the non-alcoholic fatty liver. Although the mechanism of saturated fatty acid-induced toxicity in the liver is not fully understood, oxidative stress may be deeply involved. We examined the effect of increased palmitic acid, the most common saturated fatty acid in the blood, on the liver of BALB/c mice via tail vein injection with palmitate. After 24 h, among several anti-oxidative stress response genes, only heme oxygenase-1 (HO-1) was significantly upregulated in palmitate-injected mice compared with that in vehicle-injected mice. Elevation of HO-1 mRNA was also observed in the fatty liver of high-fat-diet-fed mice. To further investigate the role of HO-1 on palmitic acid-induced oxidative stress, in vitro experiments were performed to expose palmitate to HepG2 cells. SiRNA-mediated knockdown of HO-1 significantly increased the oxidative stress induced by palmitate, whereas pre-treatment with SnCl_2_, a well-known HO-1 inducer, significantly decreased it. Moreover, SB203580, a selective p38 inhibitor, reduced HO-1 mRNA expression and increased palmitate-induced oxidative stress in HepG2 cells. These results suggest that the HO-1-mediated anti-oxidative stress compensatory reaction plays an essential role against saturated fatty acid-induced lipotoxicity in the liver.

## 1. Introduction

Non-alcoholic fatty liver disease (NAFLD) is the leading cause of liver disease worldwide [1]. Although many factors are involved in the pathogenesis [2], it is caused by the excessive accumulation of lipids in the liver [3,4], and the lipotoxicity of saturated fatty acids is considered especially important [5,6]. The most common saturated fatty acid in the blood is palmitic acid, and various mechanisms of its hepatotoxicity have been reported in vitro [7,8]. We previously reported that palmitic acid inhibited the late stage of autophagy and induced the disruption of calcium homeostasis in the endoplasmic reticulum (ER) in cultured hepatocytes [9,10]. Interestingly, when the same experiment was examined in the presence of oleic acid (the most abundant unsaturated fatty acids) to approximate in vivo conditions, the cytotoxicity of palmitic acid was almost attenuated. Based on these results, it was necessary to explore the effects of elevated blood levels of palmitic acid on the liver under in vivo conditions where various types of lipids are present. Epidemiological studies have demonstrated that NAFLD is more likely to develop when there is a high intake of palmitic acid or when there is a high level of palmitic acid in the blood and liver [11,12]. Although total free fatty acid concentrations in the blood have been reported to increase transiently in postprandial condition or by lipolysis in adipose tissue [13], there are few in vivo experiments on hepatotoxicity when palmitic acid is increased.

In addition to the effects on ER and autophagy, oxidative stress is another important factor in hepatocyte lipotoxicity, and the expression of antioxidant enzymes plays a protective role [14,15,16]. Heme oxygenase-1 (HO-1), an antioxidant enzyme that catalyzes the rate-limiting step of heme degradation, is one of the factors that protects against inflammation and oxidative stress in some conditions [17,18]. Although there are several reports on the effects of HO-1 on obesity-related pathologies [19,20], few studies have examined the role of HO-1 against the direct hepatotoxicity of palmitic acid.

In the present study, palmitate was administered directly through the tail vein of mice in order to investigate the hepatotoxicity of palmitic acid in vivo. Next, the antioxidant enzyme HO-1, which showed significant changes among the anti-oxidative stress genes, was examined for its role in palmitate-induced oxidative stress by in vitro experiments using the human hepatocyte cell line HepG2.

## 2. Materials and Methods

### 2.1. Palmitate-Injected Mice Model

Nine-week-old male BALB/c mice were purchased from Charles River Laboratories Japan (Yokohama, Japan), and all mice were fed a standard rodent chow diet and maintained under specific-pathogen-free conditions and a 12 h light/dark cycle, with ad libitum access to food and tap water. We used a modified version of a previous method for intravenous palmitate administration, which was developed for selectively increasing circulating palmitic acid levels in mice [21]. Briefly, palmitate (Tokyo Chemical Industry Co., Ltd., Tokyo, Japan) was dissolved with 1.6% lecithin (FUJIFILM Wako Pure Chemical Industries, Ltd., Osaka, Japan) and 3.3% glycerol in water to produce a mixture of 330 mM ethyl palmitate, 1.2% lecithin, and 2.5% glycerol. This mixture was then emulsified using a sonicator. Lecithin–glycerol–water solution was used as a vehicle. Furthermore, 25 μL/g (bodyweight) injection of either emulsified ethyl palmitate solution or the vehicle was slowly injected through the tail vein at a speed of 100 μL/min using a fixed winged needle. Twenty-four hours later, mice were euthanized, and tissue samples were obtained and immediately frozen. The care and handling of the animals were in accordance with the Guidelines for the Care and Use of Laboratory Animals at the Shikata Campus of Okayama University (this study was approved by the Okayama University Institutional Animal Care and Use Committee, OKU-2019313). Liver tissues of mice were fixed in 10% neutral phosphate-buffered formalin, embedded in paraffin, and sectioned. The sections were stained with hematoxylin and eosin and observed under an Olympus IX70 inverted microscope. Plasma alanine transaminase (ALT) levels were measured using FUJI DRI-CHEM 7000V (Fujifilm, Japan).

### 2.2. High-Fat-Diet-Induced Fatty Liver Mice Model

Three-week-old male C57/BL6J mice were purchased from Charles River Laboratories Japan (Yokohama, Japan). The mice were maintained under specific-pathogen-free conditions and a 12 h light/dark cycle, with ad libitum access to food and tap water. High-fat diets (HFDs) and normal diets (NDs) were purchased from Research Diets Inc. (D12492 and D12450B, respectively). At week 12, the mice were euthanized, and tissues were isolated and immediately frozen. This study was also approved by the same committee already mentioned above (OKU-2015076). Procedures for the microscopic analysis of liver tissue and measurement of plasma ALT are also described in the sections above. 

### 2.3. Cell Cultures

HepG2 was purchased from JCRB (Osaka, Japan) and maintained in Dulbecco’s modified Eagle’s medium supplemented with 10% fetal bovine serum and antibiotics (100 U/mL penicillin and 100g/mL streptomycin) at 37 °C and 5% CO_2_.

### 2.4. Preparation of Palmitate/Bovine Serum Albumin Complex Solution for Cell Culture

Palmitate (Sigma-Aldrich, St. Louis, MO, USA) was conjugated to fatty acid-free bovine serum albumin (Sigma-Aldrich) according to a previously described protocol [9]. Briefly, palmitate was conjugated to bovine serum albumin at a 5:1 M ratio and then stored at −20 °C in a freezer purged with inert gas; isopropanol was used as a control. The stock solution was thawed for 15 min at 55 °C and then added to the cell cultures in serum-free Dulbecco’s modified Eagle’s medium. The final concentration of palmitate in the medium was 400 μM.

### 2.5. Antibodies, Reagents, and siRNAs

The following antibodies were used: mouse anti-4-hydroxy-2-nonenal (4-HNE, Nikken Seil., Ltd.), rabbit anti-actin (Sigma-Aldrich), rabbit anti-HO-1, anti-phospho-p38, and anti-p38 (Cell Signaling Technology, MA, USA). SnCl_2_ (FUJIFILM Wako Pure Chemical) was dissolved in ethanol and used at a final concentration of 100 μM as a pre-treatment for 8 h before treatment with palmitate or the vehicle. Furthermore, cells were treated using tert-butyl hydroperoxide (t-BHP; FUJIFILM Wako Pure Chemical) as a positive control of lipid peroxidation. SB203580 (FUJIFILM Wako Pure Chemical), a selective inhibitor of p38 MAPK, was dissolved in dimethyl sulfoxide, and 10 μM of the solution was used for pre-treatment for 1 h before treatment with palmitate.

To knock down HO-1, MISSION^®^ siRNA (Sigma-Aldrich) was used, with a non-targeting siRNA as a negative control. Cells were seeded onto six-well plates or 35 mm dishes at 0.5 × 10^6^/well with HO-1-targeted siRNA or non-targeted siRNA using ScreenFect siRNA reagents (FUJIFILM Wako Pure Chemical), according to the manufacturer’s protocol for reverse transfection. 

### 2.6. Western Blotting

To analyze liver tissue and whole-cell lysates, equal amounts of protein were electrophoresed using sodium dodecyl sulfate–polyacrylamide gel electrophoresis. The membrane was blocked for 2 h in phosphate-buffered saline (PBS) containing 0.1% Tween-20 and 5% low fat dry milk (PBS-TM), and then incubated overnight at 4 °C in PBS containing 0.1% Tween-20 (PBST) containing primary antibodies at a 1:1500 dilution. The membrane was washed four times with PBST and incubated in PBS-TM containing a horseradish peroxidase-conjugated anti-mouse or rabbit secondary antibody (Dako, Denmark) at a 1:2000 dilution. The membrane was washed four times in PBST and incubated with a Western Lightning Ultra Chemiluminescence substrate (PerkinElmer, Waltham, MA, USA). The bands were visualized on an MXJB-PLUS medical X-ray film (KODAK) or using a LumiCube chemiluminescence analyzer (Liponics, Tokyo, Japan). The band intensity was quantified using ImageJ software (National Institute of Health, USA). 

### 2.7. Detection of Reactive Oxygen Species (ROS) Production

ROS production in HepG2 cells was observed using the redox-sensitive fluorescent dye 2,7-dichlorodihydrofluorescein diacetate (H_2_DCFDA). HepG2 cells were seeded in 35 mm glass-bottom culture dishes until they reached 60% confluency. After treatment with 400 μM palmitate or the vehicle for 12 h, cells were washed with PBS and then treated with 15 μM H_2_DCFDA in Hanks’ balanced salt solution (HBSS) for 20 min at 37 °C in the dark. Finally, images were obtained using a fluorescence microscope (BZ-9000, Keyence, Osaka, Japan) to determine the ROS content.

### 2.8. Lactate Dehydrogenase (LDH) Assay Using the Cell Culture Supernatant

The release amount of LDH into the supernatant was analyzed with the Cytotoxicity LDH Assay Kit-WST (Dojindo Molecular Technologies, Inc., Kumamoto, Japan), according to the manufacturer’s instructions. Relative absorbance to control was determined and calculated.

### 2.9. Measurement of Malondialdehyde (MDA) in the Liver and Culture Cell Homogenate

Phosphoric acid (120 mL) and thiobarbituric acid reagent (400 µL; 15% trichloroacetic acid/0.375% TBA/0.025N HCl) were added to 200 µL of liver tissue or cell homogenate, boiled for 45 min, cooled on ice, and centrifuged for 10 min at 600× *g*. The fluorescence of the supernatant was measured at 532/585 nm [22].

### 2.10. Analysis of Gene Expression by Quantitative Reverse-Transcription Polymerase Chain Reaction (qRT-PCR)

According to the manufacturer’s instructions, total RNA was isolated from the liver tissue homogenate or cell culture using ISOGEN (Nippon Gene, Tokyo, Japan). The isolated RNA pellets were dissolved in RNase-free water, and quantified using a Nano-drop 1000 (Nano-drop) via absorbance measurements at 260 and 280 nm. cDNA was prepared from 500 ng total RNA using ReverTra Ace^®^ qPCR RT Master Mix (Toyobo Co., Ltd., Tokyo, Japan) at 37 °C for 15 min, and the reaction was terminated at 98 °C for 5 min. Quantitative PCR was performed with THUNDERBIRD^®^ SYBR qPCR Mix (Toyobo Co., Ltd., Tokyo, Japan) using the Step One Plus Real-time PCR System (Applied Biosystems, CA, USA). Primers were designed by Sigma, applied to *Homo sapiens* and *Mus musculus* cDNA sequences obtained from the National Center for Biotechnology Information website (http://www.ncbi.nlm.nih.gov, accessed on 4 February 2021), and are listed in Table 1. Glyceraldehyde 3-phosphate dehydrogenase (*GAPDH* or *Gapdh*) was used as a housekeeping gene. For example, the relative *HO-1* mRNA expression was calculated using the following equation:Relative gene expression = 2^−ΔCt^ [ΔCt = Ct (*HO-1*) − Ct (*GAPDH*)],
where *GAPDH* is the internal control and Ct indicates the cycle threshold.

An identical cycle profile was used for all genes: 95 °C for 30 s + (95 °C for 10 s + 60 °C for 30 s) × 40 cycles. The mRNA levels of genes were calculated using the 2^−ΔΔCt^ method, where ΔΔCt was (Ct, target − Ct, GAPDH) in the treatment group, and −(Ct, target − Ct, GAPDH) control group.

### 2.11. Statistical Analysis

Data are expressed as the mean ± standard deviation and were analyzed using Prism software (GraphPad). Differences were compared between various treatment groups and controls using one-way analysis of variance followed by post hoc Tukey’s test; *p*-values < 0.05 were considered statistically significant.

## 3. Results

### 3.1. Effect of Palmitate Injection on Balb-c Mice Liver

Hematoxylin and eosin-stained liver sections showed that more lipid droplets were observed in the hepatocytes of palmitate-injected mice than in vehicle-injected mice (Control; Figure 1A). Although plasma ALT levels were not significantly different between the two groups, MDA, a lipid peroxidation marker, was significantly increased in the palmitate-injected group (Figure 1B,C). Among the indicated oxidative response genes, only HO-1 mRNA was significantly elevated in the livers of palmitate-injected mice compared with that in control mice (Figure 1D). Protein expression levels of HO-1 and 4-HNE were also significantly elevated in the livers of palmitate-injected mice compared with controls (Figure 1E).

### 3.2. Effect of HFD on C57/BL6J Mouse Livers

HFD feeding for 12 weeks induced fatty liver changes, as observed in the hematoxylin–eosin-stained sections (Figure 2A). Plasma ALT levels and MDA in the liver were elevated in HFD mice (Figure 2B,C). Furthermore, 4-HNE, another marker of lipid peroxidation, was increased in the liver of HFD mice compared with that in ND-fed mice (Figure 2D). Protein and mRNA expression of HO-1 in the livers of HFD mice was slightly, but significantly, increased than that in ND mice (Figure 2D,E).

### 3.3. Effect of HO-1 Knockdown Against Palmitate-Induced Oxidative Stress in HepG2 Cells

We performed siRNA-mediated HO-1 knockdown in HepG2 cells, which was confirmed by mRNA and protein expression analyses (Appendix A). The degree of fluorescence of H_2_DCFDA, a fluorogenic probe for ROS, was increased much more when treated with palmitate following knockdown for HO-1 compared with when it was treated with palmitate following transfection with control siRNA (Figure 3A). On the other hand, there was no difference between the treatment with control siRNA alone and HO-1 siRNA alone. The relationship between these four groups was similar for MDA (Figure 3B), immunoblotting for 4-HNE (Figure 3C, immunoblotting of HepG2 cells treated with t-BHP is on the right lane as a positive control), and LDH release (Figure 3D).

### 3.4. Effect of SnCl_2_ on Oxidative Stress in HepG2 Cells Treated with Palmitate

Increased fluorescence of H_2_DCFDA in HepG2 cells after treatment with palmitate was attenuated by pre-treatment with SnCl_2_ (Figure 4A). Increased expression of MDA and 4-HNE in HepG2 cells exposed to palmitate was significantly attenuated by SnCl_2_ (Figure 4B,C). Increased LDH release in HepG2 cells following treatment with palmitate remained unchained when pre-treated with SnCl_2_ (Figure 3D).

### 3.5. Involvement of p38/HO-1 Pathway in Palmitate-Induced Oxidative Stress in HepG2 Cells

Immunoblotting showed that phosphorylated p38MAPK levels were increased at 3, 6, and 12 h after treatment with 400 μM palmitate compared with non-phosphorylated p38MAPK level (Figure 5A). Increased mRNA levels of HO-1 induced by palmitate were significantly inhibited by SB203580 treatment (Figure 5B). Moreover, pre-treatment with SB203580 augmented the palmitate-induced increase in the expression of 4-HNE and relative LDH release (Figure 5C,D).

## 4. Discussion

In this study, we have showed, for the first time, that the elevation of palmitic acid levels in the blood clearly induced oxidative stress in the liver of Balb-c mice, although liver enzyme ALT, an indicator of liver dysfunction, was not elevated. Palmitic acid is originally abundant in the blood, and under physiological conditions where various other lipids are present, it is unlikely to cause serious organ damage even if its concentration increases to some extent. There is another report in which normal mice were intraperitoneally administrated with palmitate, which also did not show any significant change in ALT levels [23]. However, direct intravenous administration of palmitate in mice is certain to have some adverse effects on the liver, as well as impacts presented in previous reports such as myocardial damage and the dysfunction of pancreatic beta cells [21,24]. The details of how much and for how long elevated palmitic acid in the blood causes liver damage, as manifested by elevated ALT, is a subject for future research and needs to be analyzed through experiments involving different doses and durations. In this mouse experiment, we also found that one antioxidant enzyme, HO-1, responded to the increase in palmitic acid levels in the blood. This feature was reproduced in the liver of mice fed an HFD, and consistent with several reports that discuss the importance of HO-1 in other mouse models of fatty livers [25,26,27,28]. Therefore, to further investigate the relationship between HO-1 and oxidative stress, we performed experiments using HepG2 cells with palmitate exposure. Similar to the results of previous studies [29,30,31], we confirmed the cytoprotective role of HO-1 against the palmitate-induced oxidative stress. Interestingly, the HO-1 inducer, SnCl_2_, improved oxidative stress in hepatocytes but not LDH release, an indicator of cytotoxicity. Palmitate-induced cell death is involved in endoplasmic reticulum stress and autophagy dysfunction in addition to oxidative stress, which may suggest that HO-1 does not work sufficiently against these mechanisms [9,10,32]. To explore upstream of HO-1 induction, we also examined the p38-MAPK pathway, which has been reported as a signaling pathway involving HO-1 expression [33,34]. SB203580, a selective p38 inhibitor, significantly reduced HO-1 mRNA expression and increased palmitate-induced oxidative stress in HepG2 cells. This result indicated that the p38 pathway also might be a target against oxidative stress induced by palmitate in hepatocytes. Considering these results, the fact that a single palmitic acid administration into the blood of Balb-c mice did not increase ALT despite the oxidative stress in the liver might be due to the response of increased HO-1. In contrast, the elevation of ALT levels in mice fed an HFD for 12 weeks might mean that HO-1 could not fully respond to oxidative stress caused by sustained lipid-overload. Analysis of Bach1 mRNA expression, a transcription factor that negatively regulates HO-1 [35], showed no change in the livers of mice injected with palmitate but a significant increase in the liver of mice fed the HFD for 12 weeks (Appendix A). Inhibition of HO-1 by Bach1 might have contributed to the elevation of ALT in mice fed an HFD. Bach1 has also been reported to be involved in the pathogenesis of NAFLD by a similar mechanism [36]. Finally, one of the mechanisms of the antioxidant effect of HO-1 has been reported to be the generation of bilirubin, an antioxidant [37,38]. Therefore, we examined the effect of bilirubin on HepG2 cells treated with palmitate, although did not find any effect on oxidative stress (data not shown). The effect of HO-1 on palmitate-induced oxidative stress might be due to mechanisms other than those involving bilirubin.

In conclusion, HO-1 has an important role for palmitate-induced oxidative stress in the liver and the p38 pathway is involved. These mechanisms will be potential therapeutic targets for hepatic lipotoxicity.

## Figures and Tables

**Figure 1 nutrients-13-00993-f001:**
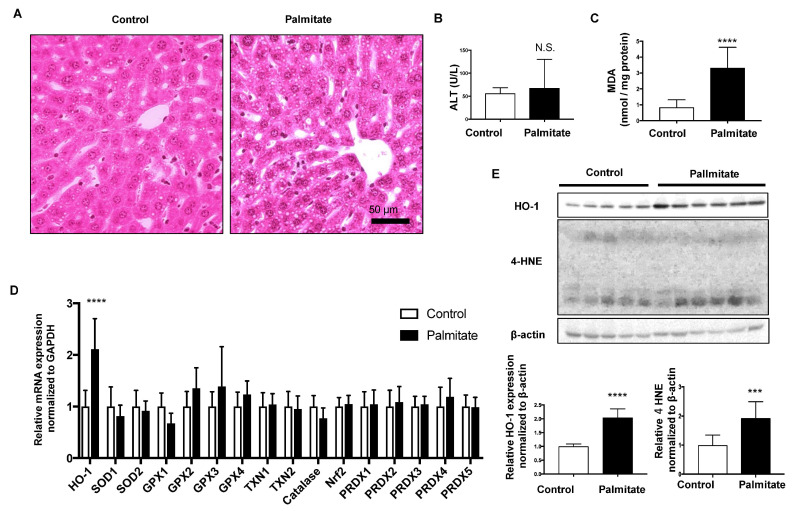
Effect of palmitate injection through the tail vein on BALB/c mice liver. Control, *n* = 10; palmitate, *n* = 11. Values are expressed as means ± SD. N.S.: not significant, *** *p* < 0.001, **** *p* < 0.0001. (**A**), Representative hepatic histological findings (HE staining: original magnification, ×20, bar = 50 μm. (**B**) The level of malondialdehyde (MDA) in mouse liver. (**C**) Plasma alanine transaminase (ALT) levels in mice. (**D**) mRNA levels of anti-oxidative stress genes in the livers of mice treated with palmitate relative to those in control mice. (**E**) HO-1 protein and 4-hydroxy-2-nonenal (4-HNE) levels in mouse livers were compared by immunoblotting. β-actin was used as a loading control. Densitometric analysis was performed.

**Figure 2 nutrients-13-00993-f002:**
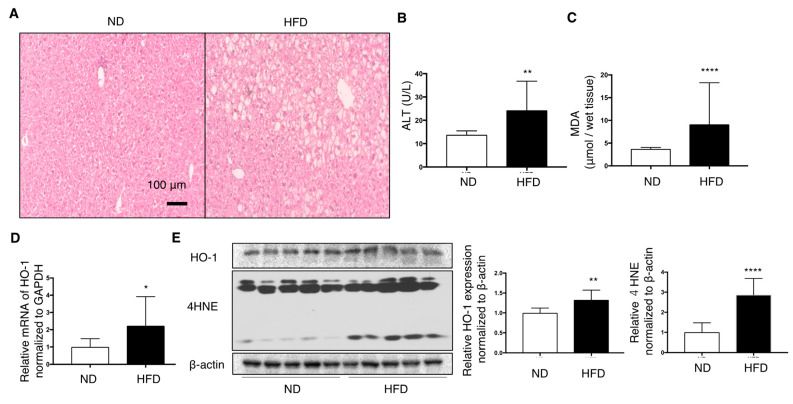
Effect of high-fat diet (HFD) on the livers of C57/BL6J mice fed for 12 weeks compared with normal diet (ND) mice (*n* = 10/group). Values are represented as means ± SD. * *p* < 0.05, ** *p* < 0.01, **** *p* < 0.0001. (**A**) Representative hepatic histological findings (HE staining: original magnification, ×20, Bar = 100 μm. (**B**) The effect of HFD on the level of malondialdehyde (MDA) in mouse livers. (**C**) Plasma alanine transaminase (ALT) levels in mice. (**D**) mRNA levels of HO-1 in the livers of mice fed a high-fat diet relative to those in ND mice. (**E**) HO-1 protein and 4-hydroxy-2-nonenal (4-HNE) levels in mice liver were measured by immunoblotting. β-actin was used as a loading control. Densitometry was performed.

**Figure 3 nutrients-13-00993-f003:**
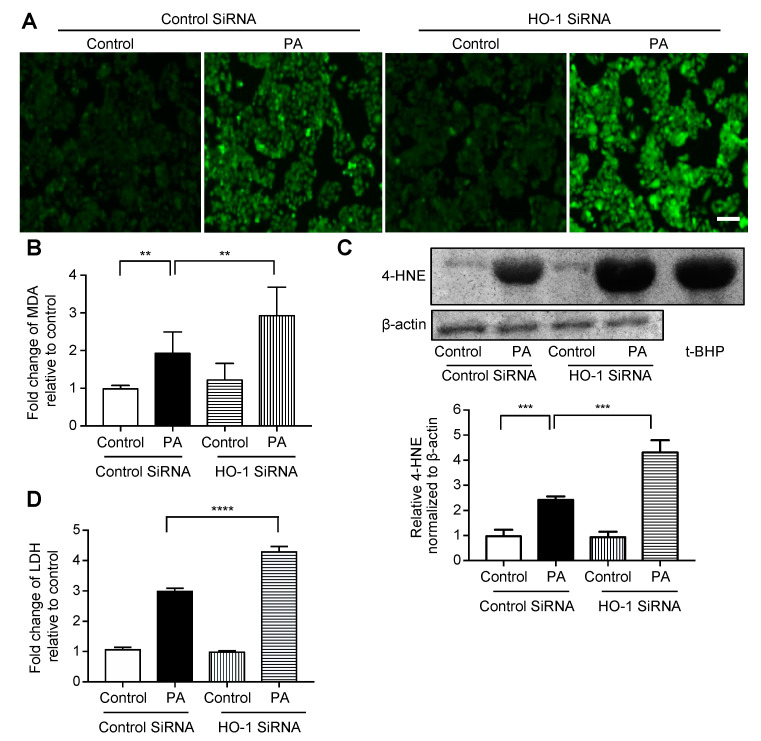
Effect of siRNA-mediated HO-1 knockdown on palmitate-induced oxidative stress in HepG2 cells. Cells were transfected with an HO-1 siRNA or control siRNA for 12 h, followed by treatment with 400 μM palmitate (PA) or vehicle (Control) for an additional 12 h. Values are expressed as means ± SD, *n* = 3. ** *p* < 0.01, *** *p* < 0.001, **** *p* < 0.0001. (**A**) After treatment with the vehicle or palmitate, cells were incubated with 5 μM of 2′,7′-dicholordihydrofluorescein diacetate (H_2_DCFDA) to detect reactive oxygen species (ROS) and were examined by fluorescence microscopy. Representative images of cells from three independent experiments are shown. Bar = 100 μm. (**B**) Relative levels of malondialdehyde (MDA) in cells. (**C**) 4-hydroxy-2-nonenal (4-HNE) levels in HepG2 cells were analyzed by immunoblotting. β-actin was used as a loading control. The positive control is presented on the right (immunoblotting of HepG2 cells treated with 100 μM of tert-butyl hydroperoxide (t-BHP) for 6 h). Densitometric analysis was performed. (**D**) Relative lactate dehydrogenase (LDH) release was calculated in the supernatant of cells.

**Figure 4 nutrients-13-00993-f004:**
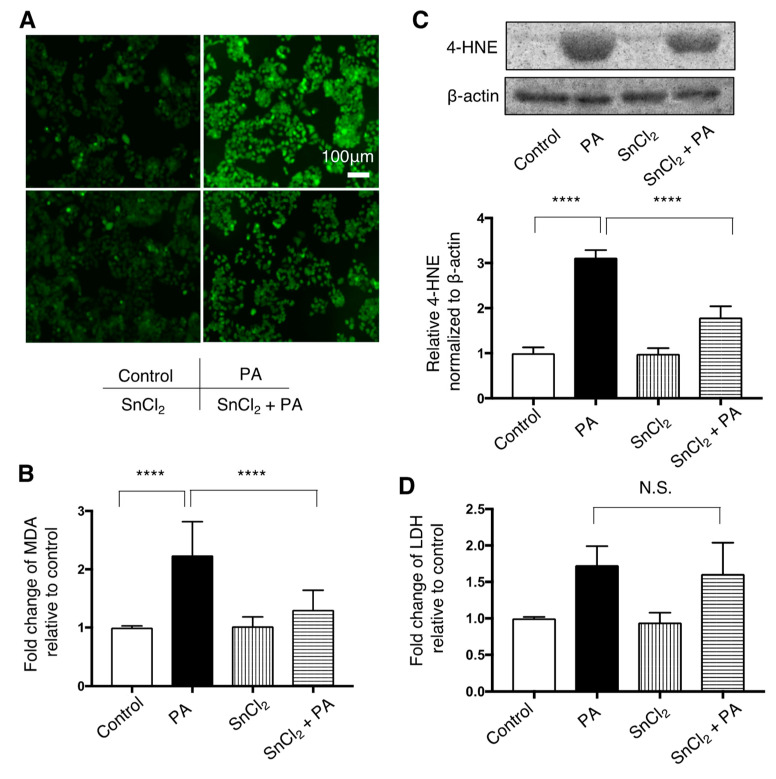
Effect of HO-1 inducer, SnCl_2_, on palmitate-induced oxidative stress in HepG2 cells. Cells were treated with a vehicle (Control) or 400 μM palmitate (PA) with or without pre-treatment with SnCl_2_ 100 μM for 8 h. Values are represented as means ± SD, *n* = 3. N.S.: not significant, **** *p* < 0.0001. (**A**) After Control or PA treatment, cells were incubated with 5 μM of 2′,7′-dicholordihydrofluorescein diacetate (H_2_DCFDA) to detect reactive oxygen species (ROS) and were observed by fluorescence microscope. Representative images from three independent experiments are shown. Bar = 50 μm (**B**) Relative levels of malondialdehyde (MDA) in cells are shown. (**C**) 4-hydroxy-2-nonenal (4-HNE) levels were compared by immunoblotting. β-actin was used as a loading control. Densitometric analysis was performed. (**D**) Relative lactate dehydrogenase (LDH) release was measured in the supernatant of HepG2 cells.

**Figure 5 nutrients-13-00993-f005:**
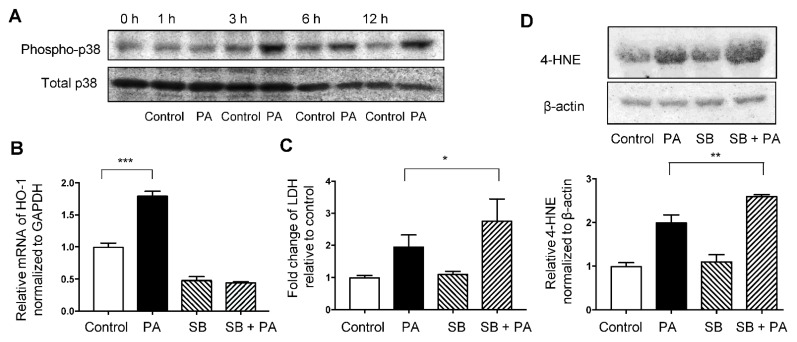
Involvement of the p38MAPK pathway in palmitate-induced oxidative stress in HepG2 cells. Values are expressed as means ± SD, *n* = 3. * *p* < 0.05, ** *p* < 0.01, *** *p* < 0.001. (**A**) Immunoblotting data of phospho-p38MAPK (Thr180/Tyr182) and total p38 in HepG2 cells treated with the vehicle (Control) or 400 μM palmitate (PA) for indicated time. (**B**) Relative HO-1 mRNA expression in HepG2 cells treated with Control or PA with or without pre-treatment of SB203580 (SB) at 10 μM for 1 h. (**C**) Relative lactate dehydrogenase (LDH) release was calculated in the supernatant of cells exposed to the Control or PA with or without pre-treatment with SB. (**D**) 4-hydroxy-2-nonenal (4-HNE) levels in HepG2 cells treated with Control or PA with or without pre-treatment with SB were analyzed by immunoblotting. β-actin was used as a loading control. Densitometric analysis was performed.

**Table 1 nutrients-13-00993-t001:** Primer sequences for gene expression analysis by quantitative real-time polymerase chain reaction.

Species	Gene	Forward (5′-3′)	Reverse (5′-3′)	Size (bp)	Accession No.
*Homo sapiens*	*HOMX1*	GGCCAGCAACAAAGTGCAAG	TGGCATAAAGCCCTACAGCA	146	NM_002133.3
	*GAPDH*	AAGGTGAAGGTCGGAGTCAA	AATGAAGGGGTCATTGATGG	108	NM_002046
*Mus musculus*	*Homx1*	CCTCACAGATGGCGTCACTT	GCTGATCTGGGGTTTCCCTC	92	NM_010442.2
	*Sod1*	GAACCATCCACTTCGAGCA	TACTGATGGACGTGGAACCC	103	NM_011434
	*Sod2*	ACTGAAGTTCAATGGTGGGG	GCTTGATAGCCTCCAGCAAC	107	NM_013671.3
	*Gpx1*	GTTTCCCGTGCAATCAGTTC	CAATGTAAAATTGGGCTCGAA	110	NM_008160.6
	*Gpx2*	CTGCAATGTCGCTTTCCCAG	CCCCAGGTCGGACATACTTG	121	NM_30677.2
	*Gpx3*	CATCCTGCCTTCTGTCCCT	ATGGTACCACTCATACCGCC	100	NM_008161.4
	*Gpx4*	GCCCACCCACTGTGGAAATG	TGGGACCATAGCGCTTCACC	130	NM_008162.4
	*Prdx1*	GCCGCTCTGTGGATGAGATT	ATCACTGCCAGGTTTCCAGC	98	NM_011034.4
	*Prdx2*	CTCCTCGGTATCTCCGCCTA	TAGCACTTGCATGACGAGCA	102	NM_007452.2
	*Prdx3*	ATGACCTACCTGTGGGACGC	GGCTTGATGGTGTCACTGC	124	NM_011563.6
	*Prdx4*	TGCCACTTCTACGCTGGTG	CCCAATAAGGTGCTGGCTTG	115	NM_016764.5
	*Prdx5*	CAGTTCTGTGCTCCGTGCAT	GCATCTCCCACCTTGATCGG	132	NM_012021.3
	*Nrf2*	TAGATGACCATGAGTCGCTTGC	GCCAAACTTGCTCCATGTCC	153	NM_010902
	*Txn1*	AGTGGATGTGGATGACTGCC	CCTTGTTAGCACCGGAGAACT	116	NM_011660.3
	*Txn2*	TGAGACACCAGTTGTTGTGGA	TTGGCGACCATCTTCTCTAGC	87	NM_019913.5
	Catalase	GATGAAGCAGTGGAAGGAGC	CCCGCGGTCATGATATTAAGT	102	NM_009804
	*Bach1*	GTCTCGGCTCCGGTCGAT	TGCTATGCACAGAGGACTCG	124	NM_007520.2
	*Gapdh*	AGGTCGGTGTGAACGGATTTG	TGTAGACCATGTAGTTGAGGTCA	123	NM_008084

*HOMX*, heme oxigenase; *GAPDH*, glyceraldehyde 3-phosphate dehydrogenase; *SOD*, superoxide dismutase; *Gpx*, glutathione peroxidase; *Prdx*, peroxiredoxin; *Nrf2*, nuclear factor, erythroid derived 2-like 2; Txn, thioredoxin; *Bach1*, BTB and CNC homology 1, basic leucine zipper transcription factor 1.

## Data Availability

Data is contained within the article or supplementary material.

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
