# Peer review of "Role of HO-1 against Saturated Fatty Acid-Induced Oxidative Stress in Hepatocytes"

_nutrients, 2021, doi:10.3390/nu13030993_

Round 1
Reviewer 1 Report
In this paper the authors aimed to establish the role for HO-1 in FA-induced oxidative stress in hepatocytes.
I do not feel that this paper present real novel data. The role of HO-1 in Palmitate induced stress has been recently establish in numerous systems including liver. I have difficulty to see the novelty in this study and do not think that its publication will add more on the subject.
Please see the following papers (some of them refer in the present study) that have been published in the last 3 years (3 in 2020) showing the implication of HO-1 in response to palmitate.
Several key papers :
Kwak HJ, Yang D, Hwang Y, Jun HS, Cheon HG. Baicalein protects rat insulinoma INS-1 cells from palmitate-induced lipotoxicity by inducing HO-1. PLoS One. 2017 Apr 26;12(4)
Shi Y, Sun Y, Sun X, Zhao H, Yao M, Hou L, Jiang L. Up-regulation of HO-1 by Nrf2 activation protects against palmitic acid-induced ROS increase in human neuroblastoma BE(2)-M17 cells. Nutr Res. 2018
Kwon CH, Sun JL, Kim MJ, Abd El-Aty AM, Jeong JH, Jung TW. Clinically confirmed DEL-1 as a myokine attenuates lipid-induced inflammation and insulin resistance in 3T3-L1 adipocytes via AMPK/HO-1- pathway. Adipocyte. 2020
Kuo NC, Huang SY, Yang CY, Shen HH, Lee YM. Involvement of HO-1 and Autophagy in the Protective Effect of Magnolol in Hepatic Steatosis-Induced NLRP3 Inflammasome Activation In Vivo and In Vitro. Antioxidants (Basel). 2020 Sep 27;9(10):924. The results indicated that silibinin could improve NASH-related oxidative stress injury by increasing the antioxidase activity and inhibiting the activities of free radical generating enzymes in the liver
Li D, Zhao D, Du J, Dong S, Aldhamin Z, Yuan X, Li W, Du H, Zhao W, Cui L, Liu L, Fu N, Nan Y. Heme oxygenase-1 alleviated non-alcoholic fatty liver disease via suppressing ROS-dependent endoplasmic reticulum stress. Life Sci. 2020 Jul 15;253:117678. doi: 10.1016/j.lfs.2020.117678. Epub 2020.
Reviewer 2 Report
This manuscript by Ogino and coauthors investigated the role of HO-1 against palmitate-induced oxidative stress in hepatocytes. They tested the direct toxicity of palmitic acid and found that only HO-1 was upregulated in the liver of palmitate-injected mice model vs. vehicle-injected mice model. They also explored the level of HO-1 mRNA both in the liver of high-fat-diet-fed mice model and HepG2 cells. Further, they studied the effect of HO-1 on palmitate-induced oxidative stress in HepG2 cells. The experimental design is reasonable. The data presented support the conclusion drawn, which is useful for others in the related research field. In view of this, the manuscript may be accepted for publication after minor revise.
Please consider addressing the following comments:
- The research background and rationale presented in the paper is not well described and poor informative.
- Why do authors use male mice, but not half male and half female mice? Is gender affect the result?
Reviewer 3 Report
The manuscript by Ogino et al. discusses the role of HO-1 in palmitic acid-induced oxidative stress in hepatic cells. In my opinion, both the introduction chapter and the discussion are too general and need to be developed in the context of discussed issues.
The figures and results are well compiled, however the minor remarks relate to: Fig.1 do the authors have a better quality of WB image?
The abbreviations used in the graphs should be explained in the description below the Figure, e.g. Fig. 3 t-BHP.
Round 2
Reviewer 1 Report
The authors answers some of my criticism about the novelty.